# β-Cyclocitral Does Not Contribute to Singlet Oxygen-Signalling in Algae, but May Down-Regulate Chlorophyll Synthesis

**DOI:** 10.3390/plants11162155

**Published:** 2022-08-19

**Authors:** Thomas Roach, Theresa Baur, Ilse Kranner

**Affiliations:** Department of Botany, University of Innsbruck, Sternwartestraße 15, 6020 Innsbruck, Austria

**Keywords:** high light stress, singlet oxygen, signalling, GPX5, beta cyclocitral, acrolein, glutathione peroxidase, carbonyl, transcription

## Abstract

Light stress signalling in algae and plants is partially orchestrated by singlet oxygen (^1^O_2_), a reactive oxygen species (ROS) that causes significant damage within the chloroplast, such as lipid peroxidation. In the vicinity of the photosystem II reaction centre, a major source of ^1^O_2_, are two β-carotene molecules that quench ^1^O_2_ to ground-state oxygen. ^1^O_2_ can oxidise β-carotene to release β-cyclocitral, which has emerged as a ^1^O_2_-mediated stress signal in the plant *Arabidopsis thaliana*. We investigated if β-cyclocitral can have similar retrograde signalling properties in the unicellular alga *Chlamydomonas reinhardtii*. Using RNA-Seq, we show that genes up-regulated in response to exogenous β-cyclocitral included *CAROTENOID CLEAVAGE DIOXYGENASE 8* (*CCD8*), while down-regulated genes included those associated with porphyrin and chlorophyll anabolism, such as tetrapyrrole-binding protein (*GUN4*), magnesium chelatases (*CHLI1*, *CHLI2*, *CHLD*, *CHLH1*), light-dependent protochlorophyllide reductase (*POR1*), copper target 1 protein (*CTH1*), and coproporphyrinogen III oxidase (*CPX1*). Down-regulation of this pathway has also been shown in β-cyclocitral-treated *A. thaliana*, indicating conservation of this signalling mechanism in plants. However, in contrast to *A. thaliana*, a very limited overlap in differential gene expression was found in β-cyclocitral-treated and ^1^O_2_-treated *C. reinhardtii*. Furthermore, exogenous treatment with β-cyclocitral did not induce tolerance to ^1^O_2_. We conclude that while β-cyclocitral may down-regulate chlorophyll synthesis, it does not seem to contribute to ^1^O_2_-mediated high light stress signalling in algae.

## 1. Introduction

Photosynthetic organisms often encounter suboptimal conditions, leading to the absorption of excess light energy and ‘high light’ (HL) stress. Therefore, light harvesting must be regulated, requiring acclimation to the current environment [1]. As part of acclimation, signals from the chloroplast can alter transcription in the nucleus in so-called retrograde signalling [2,3]. For example, levels of tetrapyrrole intermediates (chlorophyll precursors) in the chloroplast provide feedback cues to the nucleus during chlorophyll synthesis [4,5]. Stress signalling is also partially orchestrated by reactive oxygen species (ROS), including singlet oxygen (^1^O_2_), which is produced by energy transfer from excited chlorophyll to molecular oxygen in photosystem II (PSII) [6,7,8]. Singlet oxygen oxidises almost anything in its path and the ^1^O_2_ signal leaves the chloroplast in the form of down-stream reaction products. β-cyclocitral, a ^1^O_2_-derived breakdown product of β-carotene, has emerged as an aldehyde electrophile involved in ^1^O_2_ retrograde signalling of *Arabidopsis thaliana* [9]. However, the contribution of β-cyclocitral to ^1^O_2_ signalling in algae is unknown.

Other ^1^O_2_-derived molecules with potent signalling activity include α,β-unsaturated carbonyl derivatives, known as reactive carbonyl/electrophile species (RES). These are produced as a consequence of lipid peroxidation [10,11]. Thylakoid membranes are particularly enriched in 1-linolenoyl/2-linolenoyl (di-Cl8:3) found in monogalactosyldiacylglycerol, a polar lipid that improves fluidity for membrane functionality, but is highly prone to peroxidation by ^1^O_2_. One of the most abundant RES produced by lipid peroxidation in chloroplasts due to HL stress is acrolein [10,11], which can activate a significant proportion of transcriptional changes that occur in response to ^1^O_2_ [12,13]. In *Chlamydomonas reinhardtii*, an electrophile response element (ERE)-containing bZIP transcription factor called SOR1 participates in ^1^O_2_ signalling [12,13]. SOR1 up-regulates transcription of a large suite of genes, including glutathione transferases (e.g., *GSTS1*) and an isoflavone reductase-like protein (*IRL1*), which contribute to RES-associated antioxidant defences [12,14,15].

Although they are potent signalling molecules, RES are also toxic to cells by forming Michael addition adducts with nucleophilic thiolate anions [16], such as redox-active cysteine residues of proteins. As a thiol, the antioxidant glutathione (GSH), can detoxify RES. *Chlamydomonas reinhardtii* rapidly responds to low concentrations of the RES acrolein (≤600 ppm) by increasing GSH contents, but is critically GSH-depleted at higher acrolein concentrations [13]. Another component of RES detoxification putatively includes glutathione peroxidases (GPX), whereby *GPX5* (also known as GPXh) transcription is strongly up-regulated by sub-lethal levels of ^1^O_2_ [17,18] and acrolein [13]. Moreover, overexpression of GPX5 in *C. reinhardtii* can increase tolerance to ^1^O_2_ [18].

In summary, low levels of ROS/RES can activate signalling pathways implicated in acclimation, whereas an excess ROS/RES load leads to intolerable stress. Thus, stress responses can be distinguished into eustress and distress: Eustress leads to increased stress tolerance via acclimation in which ^1^O_2_ signalling can be involved, and distress leads to loss of viability due to excess stress (e.g., high RES load) beyond a level that can be compensated for by acclimation [18,19,20].

Here, we explored the potential role for β-cyclocitral in ^1^O_2_-mediated signalling and inducing ^1^O_2_ tolerance in the unicellular model green alga, *C. reinhardtii*. First, we confirmed that β-cyclocitral could enter cells by measuring increased concentrations of the molecule in treated cells, and observing the concentration-dependent effect on chlorophyll fluorescence. Then, using RNA-Seq analysis, we analysed the transcriptional response of cells to β-cyclocitral and compared this to previously published data of differential gene expression induced by the photosensitizer rose bengal (RB), to reveal if elements of ^1^O_2_ signalling were activated by β-cyclocitral. Further experiments investigated if β-cyclocitral can induce ^1^O_2_ tolerance, and comparisons are drawn with responses to acrolein, a ^1^O_2_-derived signal resulting from lipid peroxidation of the thylakoid membrane.

## 2. Results

As an aldehyde, β-cyclocitral could be cytotoxic as well as a signalling molecule. Therefore, to assess toxicity, the impact of various concentrations of β-cyclocitral on photosynthesis was probed via chlorophyll fluorescence. The maximum quantum yield of PSII (*F*_v_/*F*_m_), which is an often-used health marker of photosynthetic organisms, decreased in both wild types (WT), cell-wall containing WT-4A and cell-wall-less cw15, in response to β-cyclocitral treatment with ≥10 μL/Petri dish (Figure 1A), corresponding to ≥50,000 ppm atmospheric concentration (see methods for calculation). In contrast, NPQ that is also measured via chlorophyll fluorescence was affected at much lower β-cyclocitral concentrations (Figure 1B), including at 0.12 μL/Petri dish (Appendix A), which corresponds to the 600 ppm treatment used for the RNA-Seq analysis. Cellular concentrations of β-cyclocitral before treatment were 0.2 nmol g^−1^ fresh weight, and increased 400 fold 2 h after exogenous treatment at 600 ppm (Appendix A). There was no difference in the influence of β-cyclocitral on NPQ or *F*_v_/*F*_m_ between WT-4A and cw15 strains (Figure 1). Furthermore, the reduction of NPQ was also found in the *npq4*, *stt7,* and *npq4stt7* mutants (Appendix A), and thus was independent of LHCSR3- and STT7-mediated NPQ, which are the major NPQ mechanisms in *C. reinhardtii* [1,21]. Therefore, our results are indicative of a direct physical effect of β-cyclocitral on NPQ that occurs at the level of the thylakoid membrane, similar to an uncoupler. To test this, the activity of the pH-dependent violaxanthin cycle was measured in response to HL. Less zeaxanthin accumulated in β-cyclocitral-treated cells (Appendix A), indicating that the proton gradient was dissipated by β-cyclocitral.

Aldehydes can be detoxified by GSH and associated enzymes, such as GSTS1. Thus, the response of β-cyclocitral on GSH concentrations was also measured up to 1500 ppm (0.3 μL/Petri dish), and after 4 h, no effect was observed (Appendix A).

Overall, the lack of decrease in *F*_v_/*F*_m_ and lack of change in GSH concentrations indicated that β-cyclocitral was not a cytotoxic aldehyde at treatment with up to 1500 ppm, a concentration at which the RES acrolein is lethal [13]. Moreover, the significant impact of β-cyclocitral on NPQ in WT-4A and cw15 showed that this aldehyde could enter the chloroplast of *C. reinhardtii* and that the cell wall was not a hindrance to influx. Therefore, we felt confident to be able to assess the signalling properties of β-cyclocitral in exogenously-treated cells.

One of the most ^1^O_2_-responsive genes in *C. reinhardtii* is *GPX5*, and GPXs likely have a role in mitigating HL stress by detoxifying aldehydes/RES that are produced as a consequence of ^1^O_2_ production [18]. We screened three mutants over-expressing *GPX5* in the WT-4A background [18] and found clearly elevated protein levels of GPX5 in *GPXHOX-11* and *GPX5OX-14*, relative to WT-4A, in low light (LL)-treated cells (Figure 2A). This led to the selection of *GPXHOX-11* for further experiments to see how β-cyclocitral, acrolein, and HL influence GPX5 levels, and how elevated levels of GPX5 affect the response to these treatments and subsequent tolerance to ^1^O_2_. Levels of GPX5 increased in WT-4A cells after 4 h of HL, indicating that this treatment induced ^1^O_2_ production. However, 4 h treatments with 600 ppm of β-cyclocitral or 600 ppm of acrolein did not increase GPX5 levels (Figure 2B). After 4 h of HL, WT cells increased the GSH contents, indicating that cells were under mild oxidative stress, whereas in *GPXHOX-11,* no change in the GSH contents occurred, indicating that cells were under less stress. In support of this, after 4 h of HL, *GPXHOX-11* accumulated less aldehydes (propanal and hexanal) and RES (acrolein and 4-hydroxynonenal) than WT-4A, whereas the fold change of β-cyclocitral was equally low in both genotypes (Figure 3). After a 4 h treatment with 600 ppm of acrolein, both WT and *GPXHOX-11* accumulated GSH (Appendix A).

A function of ^1^O_2_-related signalling is partly towards increasing tolerance of this ROS [18]. Therefore, the influence of exogenous β-cyclocitral and acrolein treatments under LL (as potential components of ^1^O_2_-mediated signalling) on the tolerance to ^1^O_2_ was measured. The influence of HL stress, which increases endogenous ^1^O_2_ levels, was also included as pre-treatment before testing ^1^O_2_ tolerance, and all comparisons were made to LL-treated ‘control’ cells. Tolerance to ^1^O_2_ was tested by incubating cells with the photosensitizer RB. Since the amount of ^1^O_2_ that RB produces is dependent on the degree of photoexcitation (i.e., intensity and duration of light treatment) and RB concentration, various treatments were conducted to test cell tolerance. These constituted 4 μM, 7 μM, and 10 μM of RB, either for 10 min at 250 μmol photons m^−2^ s^−1^ to provide a short ^1^O_2_ shock, or 24 h at 50 μmol photons m^−2^ s^−1^ to test longer term endurance. The 24 h endurance test was more severe and led to less cell survival of control cells, under which *GPXHOX-11* was significantly less affected than WT-4A (Figure 4), with *p* < 0.05 when comparing genotype as a factor with MANOVA across all RB concentrations. Pre-treatment with HL increased the tolerance of WT-4A to the ^1^O_2_ shock treatment with 7 μM and 10 μM of RB (Figure 4A), and the 24 h treatment with 4 μM (Figure 4B), while affecting *GPXHOX-11* less. This resulted in HL-treated WT-4A and *GPXHOX-11* having a similar ^1^O_2_ tolerance (Figure 4). Relative to control cells, pre-treatment with 600 ppm of β-cyclocitral had no impact on ^1^O_2_ tolerance in either genotype under all treatments (average *p* = 0.895), whereas pre-treatment with 600 ppm of acrolein increased the tolerance of both genotypes (average *p* = 0.034). In summary, acrolein and HL induced tolerance to severe ^1^O_2_ stress, whereas β-cyclocitral did not significantly affect tolerance.

To investigate if β-cyclocitral could contribute to ^1^O_2_-mediated signalling in *C. reinhardtii*, an RNA-Seq analysis of cells treated for 2 h with 600 ppm of β-cyclocitral under LL (Appendix A) was conducted and compared to the transcriptional response of cells treated with the ^1^O_2_-producing photosensitizer RB at 1 μM under LL (data from [13]). In response to the β-cyclocitral treatment, only six genes were significantly up-regulated, and 57 genes were down-regulated, when considering a fold change of >2 and modified *t*-test *p* values of <0.01 (Appendix A). Of the genes down-regulated by β-cyclocitral, 18 and 6 were significantly down-regulated and up-regulated, respectively, by RB (Appendix A; Appendix A). Differentially expressed genes associated with carotenoid metabolism include carotenoid cleavage dioxygenase 8 (*Cre08.g365851*), up-regulated five-fold, and β-carotene ketolase (*BKT1*), up-regulated two-fold, but not significantly (Appendix A). Of all significantly down-regulated genes, the only ontological group with >1 hit was ‘porphyrin and chlorophyll metabolism’ with 10 hits: *Cre01.g015350* (*POR1*), *Cre01.g050950*, *Cre02.g085450* (*CPX1*), *Cre05.g242000* (*CHLD*), *Cre05.g246800* (*GUN4*), Cre06*.g306300* (*CHLI1*), *Cre07.g325500* (*CHLH1*), *Cre09.g396300* (*PPX1*), *Cre12.g510050* (*CTH1*), *Cre12.g510800* (*CHLI2*), and *Cre16.g663900*. Collectively, these genes covered several steps of chlorophyll anabolism (Appendix A), but the overall overlap of differential gene expression with RB-treated cells was low, as shown by a R^2^ linear correlation of <0.01 when including all genes, which increases to 0.15, considering only the 63 genes with significantly altered expression (Figure 5). In comparison, this contrasts with the much tighter relationship between differential gene expression shared between acrolein-treated and RB-treated cells [13], which has an R^2^ linear correlation of 0.34 and 0.70 when considering expression of total genes and only significantly affected genes, respectively (Figure 5).

## 3. Discussion

Ten years ago, the discovery that β-cyclocitral in *A. thaliana* effects transcription of several genes know to be affected by ^1^O_2_ [9] made a coherent link between HL-induced ^1^O_2_ production and ROS-associated retrograde signalling. Subsequently, it was found that β-cyclocitral functions up-stream of MBS1 [22], a zinc finger protein that regulates ^1^O_2_-dependent gene expression, not only in *A. thaliana* but also in *C. reinhardtii* [23]. However, the fact of whether β-cyclocitral actually has a role in ^1^O_2_ signalling in alga remained unknown. Since then, other RES (i.e., acrolein), related to lipid peroxidation rather than carotenoid cleavage, emerged as retrograde signals acting in ^1^O_2_-mediated stress acclimation [13]. Here, we investigated how β-cyclocitral modulates the physiology and transcription in *C. reinhardtii* and made comparisons with transcriptional responses of cells to the photosensitizer RB and the RES acrolein.

The very minor effect of β-cyclocitral on *F*_v_/*F*_m_ showed how tolerant cells were of this molecule. For example, decreases in *F*_v_/*F*_m_ occurred at >1000 fold concentrations compared to the effects of acrolein (Figure 1A; [13]). The concentration of β-cyclocitral even in very light-stressed *C. reinhardtii* has never been measured at >1 nmol g^−1^ fresh weight [13,24], which is below the cellular concentrations after exogenous treatment with 600 ppm (Appendix A). Therefore, our data supports that, unlike acrolein, β-cyclocitral does not build up to toxic concentrations in light-stressed cells. Despite the insensitivity of *F*_v_/*F*_m_ to β-cyclocitral, NPQ was affected at much lower concentrations (Figure 1B and Appendix A), confirming that β-cyclocitral was able to enter cells, which otherwise may have contributed to tolerance. Inhibition of NPQ in various NPQ mutants indicates that β-cyclocitral directly affected an over-riding NPQ mechanism, such as the requirement of low luminal pH. This was indeed shown by the lower accumulation of zeaxanthin under HL, which is a process requiring a low luminal pH for violaxanthin de-epoxidase activity, in β-cyclocitral-treated cells (Appendix A). Therefore, we suggest that β-cyclocitral may act as an uncoupler of the thylakoid membrane potential, comparable to the inhibitory activity of structurally similar monoterpene ketones, such as pulegone, on respiration [25]. The relevance of this observation is that NPQ protects from ^1^O_2_ production under HL [1,24], and thus exogenous β-cyclocitral may increase ^1^O_2_ production and may confound observations of β-cyclocitral involvement in ^1^O_2_ signalling under HL. Here, exogenous treatments for the RNA-Seq were conducted under very LL (2 μmol photons m^−2^ s^−1^), thus unaffected by lowered NPQ.

Defence against ^1^O_2_ requires many enzymes that mitigate lipid peroxidation, including GPX5 [18]. In mammalian cells, GSH is the typical GPX substrate to break down H_2_O_2_, whereas in *C. reinhardtii,* a thioredoxin is the reductant of GPX5 that has a close association with ^1^O_2_ stress [26]. In WT cells, HL induced accumulation of GPX5 (Figure 2B), alongside higher levels of RES, which were attenuated in *GPXHOX-11* (Figure 3), a *GPX5* over-expressing mutant with elevated GPX5 levels (Figure 2). This supports that GPX5 has a role in metabolising HL-induced RES production, and explains why HL stress is associated with elevated GPX5 levels [26,27]. *GPXHOX-11* also possessed significantly elevated tolerance to long-term ^1^O_2_ treatment with the photosensitizer RB (Figure 4B), in agreement with results from Ledford et al. [18]. Comparing the influence of pre-treatments on ^1^O_2_ tolerance, acrolein was able to induce tolerance of WT and *GPXHOX-11*, whereas β-cyclocitral could not (Figure 4). Β-cyclocitral did not induce GSH synthesis, whereas acrolein and HL did (Appendix A), as also previously shown [13]. Thus, enhanced ^1^O_2_ tolerance can be partially attributed to elevated GSH and GPX5 levels.

A molecule involved in stress signalling would be expected to increase in concentration in response to the relevant stress. The amounts of β-cyclocitral in *C. reinhardtii* were ≤1 nmol g^−1^ fresh weight (Appendix A), and not affected by HL stress (Appendix A and Figure 3), also in agreement with previous data [13,24]. In comparison, levels of RES increased in HL-stressed cells, similar to after treatment with RB [13], supporting that ^1^O_2_ production under HL is involved in RES production, but hardly with β-cyclocitral production. Acrolein has received attention in the field of redox biology for its high electrophilic nature and high endogenous levels of >5 nmol g^−1^ fresh weight in stressed plants and algae alike [13,28,29]. Previously, we showed that exogenous acrolein treatments induce a ‘eustress’ (i.e., acclimation) response by up-regulating thiol-disulfide-dependent defence mechanisms required for tolerating ^1^O_2_ [13]. Of note, around half of global gene expression (up and down) occurring in response to 600 ppm of acrolein, the dose that induced a eustress response with highest tolerance to ^1^O_2_, was shared with the gene regulation in response to RB [13]. While we did not find a similar transcriptional response to β-cyclocitral (Figure 5), in line with a lack of inducing ^1^O_2_ tolerance (Figure 4), there was evidence that β-cyclocitral may have some signalling properties in C*. reinhardtii*. Collectively down-regulated genes (Appendix A) covered many steps of chlorophyll anabolism (Appendix A), such as porphobilinogen deaminase/HemC (*Cre16.g663900.t1.2*), coproporphyrinogen III oxidase (*CPX1*), and protoporphyrinogen oxidase (*PPX1*), which are involved in early steps of porphyrin synthesis, as well as genes coding for proteins that insert Mg^2+^ into protoporphyrin, including tetrapyrrole-binding protein (*GUN4*) and magnesium chelatase (*CHLI1*, *CHLI2*, *CHLD*, *CHLH1*) to form Mg-protoporphyrin IX (MgP), the first dedicated intermediate of the chlorophyll branch. In *C. reinhardtii*, *CHLI2* seems to be redundant to *CHLI1* [30]. Gene expression associated with later steps of chlorophyll synthesis, such as copper target 1 protein (*CTH1*) and light-dependent protochlorophyllide reductase (*POR1*) were also down-regulated by β-cyclocitral. In *A. thaliana*, β-cyclocitral also down-regulated expression of *CHLI2* and a few other genes involved in porphyrin/chlorophyll biosynthesis, such as *PORB*, *CHLM* and *HEME1* alongside an up-regulation of *CLH1* and *CLH2* involved in chlorophyll catabolism (Ramel et al., 2102), indicating a conserved signalling role for β-cyclocitral in decreasing chlorophyll contents, which existed before the evolution of vascular plants. Chlorophyll synthesis needs to be tightly regulated because MgP and protochlorophyllide are, similar to free chlorophyll, highly efficient photosensitizers [6]. MgP provides feedback on chlorophyll synthesis by repressing nuclear transcription in a signalling pathway that requires GUN4 [4]. There are contrasting reports on whether the GUN4-MgP complex produces more or less ^1^O_2_ than MgP alone, and if ^1^O_2_ is a component of the retrograde signal [31,32].

In summary, in *C. reinhardtii,* β-cyclocitral does not seem to have a role in ^1^O_2_ signalling or inducing ^1^O_2_ tolerance. Nonetheless, the influence of carotenoid cleavage products on chlorophyll synthesis and how ^1^O_2_ integrates into the retrograde signalling of this pathway warrants further investigation.

## 4. Materials and Methods

### 4.1. Strains and Growth Conditions

*Chlamydomonas reinhardtii* WT-4A^+^ (CC-4051) and *GPXHOX*^+^ strains *GPXHOX-10* (CC-4606), *GPXHOX-11* (CC-4607), and *GPXHOX-14* (CC-4608) in the WT-4A^+^ background, from Ledford et al., [18], were initiated in Tris-Acetate-Phosphate (TAP) liquid media, pH 7.0. For agar-grown cultures, 1 mL of liquid culture was evenly spread across 11 cm Petri dishes half-filled with 1.5% (*w*:*v*) TAP agar media. The liquid medium was evaporated for 0.5 h in a sterile air-flow bench before the lid was replaced, but not sealed. Liquid and agar-grown cultures were grown under constant LL (50 μmol photons m^−2^ s^−1^) at 20 °C in a growth chamber (Percival PGC-6HO, CLF Plant Climatics GmbH, Wertingen, Germany). HL was provided by increasing the light intensity of the growth chamber to 600 μmol photons m^−2^ s^−1^ for 4 h.

### 4.2. Chlorophyll Fluorescence Measurements

Pulse-amplitude modulation (PAM) measurements of chlorophyll fluorescence (*F*) parameters were performed with an Imaging PAM (WALZ). After dark treatment, minimum (*F*_o_) and maximum fluorescence (*F*_m_) was measured immediately before and during a 200 ms saturating pulse (6000 μmol photons m^−2^ s^−1^), respectively. The maximum quantum yield of photosystem II (*F*_v_/*F*_m_) was calculated via (*F*_m_-*F*_o_)/*F*_m_ after 1.5 h recovery in dark. NPQ was calculated via (*F*_m_- *F*_m_′)/*F*_m_, with *F*_m_′ measured after 2 min at 750 μmol photons m^−2^ s^−1^.

### 4.3. Exogenous β-Cyclocitral Treatments

For treatments, advantage was taken of the volatility of β-cyclocitral (Sigma-Aldrich), which was placed on a paper wick within the middle of a sealed Petri dish to treat agar-grown cells, in an identical approach used for acrolein treatments [13]. Β-cyclocitral was diluted in 100% p.a. methanol and 1 μL containing 0–0.3 μL of diluted β-cyclocitral (mock = 1 μL pure methanol) was placed on a paper wick in an Eppendorf lid in the centre of a 11 cm Petri dish, which was immediately sealed with Parafilm and left under very LL (2 μmol photons m^−2^ s^−1^). After 2–4 h, as indicated, cells within 2 cm of the middle of the plate were gently scraped from the agar with a spatula and immediately frozen in liquid nitrogen prior to biochemical analyses. Β-cyclocitral concentrations in parts per million (ppm) were calculated on a volume basis, considering a 30 cm^3^ air space in a 11 cm Petri dish, a β-cyclocitral density of 0.943 g mL^−1^, 95% purity, and the particle-related gas concentration of 0.0241 m³/mol, so that 0.12 μL of β-cyclocitral corresponded to 600 ppm inside the Petri dish.

### 4.4. HPLC Analysis of Glutathione and Pigments, Western Blotting of Glutathione Peroxidase 5, and LC-MS/MS Measurement of RES

All methods were conducted according to Roach et al., [13], after 4 h treatment. For HPLC analyses (glutathione and pigments), cells were first freeze-dried for 3 days and each replicate was composed of 8–10 mg dry weight. For Western blotting of GPX5 levels and LC-MS/MS analyses of RES, cells were not freeze-dried. Loading of protein extracts and normalisation of pigments was made to total chlorophyll of the extract, according to [13]. For western blotting, the GPXh antibody (AS15 2882, Agrisera, Vännäs, Sweden) at a ratio of 1:10.000 and PsbA antibody (AS05 084, Agrisera, Vännäs, Sweden) at a ratio of 1:25.000 were used.

### 4.5. Singlet Oxygen Resistance Test

Resistance to singlet oxygen was performed by suspending liquid cultures in fresh TAP media with RB at 0, 4, 7, and 10 μM of RB in a 96-well multiwall plate with a total volume of 200 μL per well. One light treatment consisted of a ^1^O_2_ shock by exposure to 250 μmol photons m^−2^ s^−1^ for 10 min before recovery at 20 μmol photons m^−2^ s^−1^, and another was constant exposure to 50 μmol photons m^−2^ s^−1^. After 24 h, cell density at 650 nm was measured as an indicator of cell number for calculating differences between each RB treatment and cells without RB for each genotype and pre-treatment individually.

### 4.6. RNA-Seq Analysis

Total RNA was extracted with the Rneasy Plant Mini Kit (Qiagen, Hilden, Germany) and additional on-column Dnase treatment (Rnase-free Dnase set, Qiagen) from ca. 15 mg (fresh weight) per replicate (=1 Petri dish). Cells were harvested after a 2 h treatment with 600 ppm of β-cyclocitral diluted in methanol, or for control with methanol only. Poly A-enriched library preps were sequenced with an Illumina HiSeq2500 using single-end 50 bp read lengths, by the NGS Core Facility of the Vienna Biocentre, Austria, resulting in an average of 24,063,263 reads per replicate, *n* = 3. Reads were aligned against the *C. reinhardtii* reference genome (JGI v5.5 release) with STAR version 2.5.1b, created by Dobin et al. [33] (Cold Springs Harbour, New York, NY, USA), using 2-pass alignment mode.

### 4.7. Data Analysis and Statistics

For all measurements, one Petri dish of cell culture counted as an individual biological replicate. The RNA-Seq analysis data was analysed by the Bioinformatics and Scientific Computing Core of the Vienna Biocenter Core Facilities with the R package limma for selecting genes that were differently expressed more than two-fold between control and treatment (i.e., mock cells v β-cyclocitral-treated cells). The *p*-values used for filtering differentially expressed genes were cut off at *p* < 0.01. KEGG pathways and gene ontology annotations (*Chlamydomonas*-based) were conducted using the Algal Functional Annotation Tool [34].

Significant differences for biochemical measurements and chlorophyll fluorescence at *p* < 0.05 were calculated in IBM SPSS Statistics, version 24, IBM, (New York, NY, USA) using one-way ANOVA with Tukey’s post hoc test, or for pairwise comparisons using t-test with independent samples. A multivariate general linear model (MANOVA) was additionally calculated to evaluate the influence of genotype on RES tolerance.

## 5. Conclusions

Overall, we first conclude that β-cyclocitral is not a particularly reactive aldehyde since it does not lead to loss of *F*_v_/*F*_m_ or modulate GSH concentrations like RES do. Second, unlike most RES, β-cyclocitral concentrations are not associated with HL stress in *C. reinhardtii*. Third, GPX5 helps mitigate HL stress by breaking down RES produced by ^1^O_2_. Fourth, β-cyclocitral is unable to induce tolerance to ^1^O_2_, and does not seem to contribute to ^1^O_2_ signalling in *C. reinhardtii*, but instead may have a specific role in down-regulating chlorophyll synthesis.

## Figures and Tables

**Figure 1 plants-11-02155-f001:**
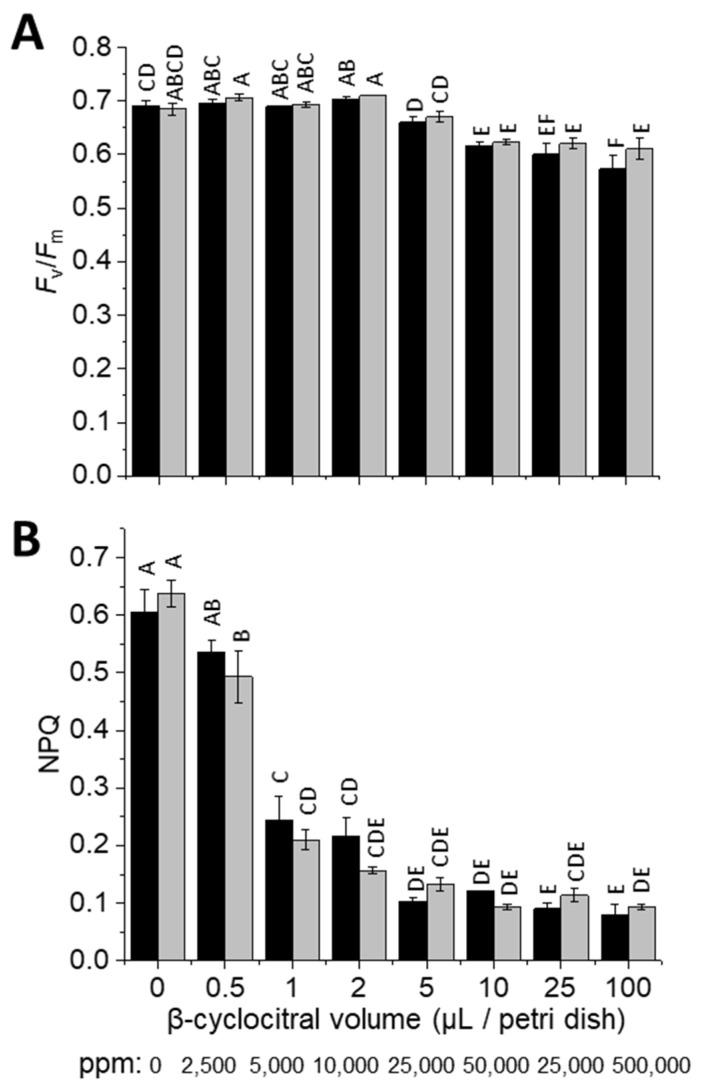
Effect of exogenous β-cyclocitral on *F*_v_/*F*_m_ and NPQ in *Chlamydomonas reinhardtii.* (**A**) Measurements of *F*_v_/*F*_m_ and (**B**) NPQ in 4 h HL-acclimated cultures were made after 4 h of treatment under very LL (see methods), of cell-wall-containing wild type (WT-4A; black bars) and cell-wall-less (cw15; grey bars) cultures, *n* = 3 ± SD, with distinct letters indicating significant differences (*p* < 0.05).

**Figure 2 plants-11-02155-f002:**
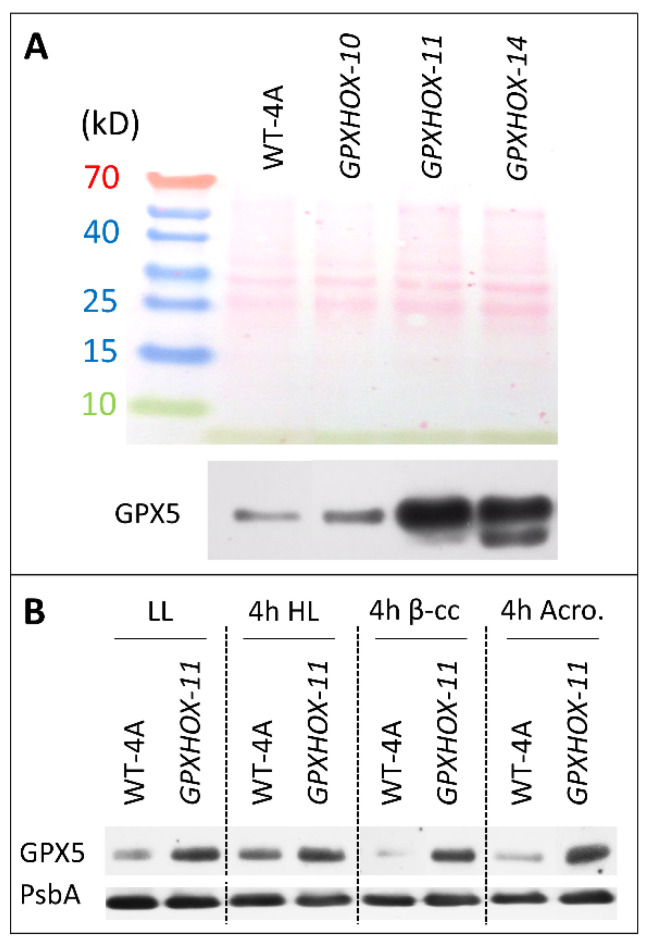
Protein levels of GPX5 in WT-4A and *GPX5*-overexpressor (*GPXHOX)* mutants under LL and in response to HL, β-cyclocitral and acrolein. (**A**) Three *GPXHOX* lines and WT-4A were analysed for GPX5 protein levels under LL. Shown above is the Ponceau-stained membrane for loading control. (**B**) The effect of LL, 4 h with HL, and 4 h with 600 ppm of β-cyclocitral (β-cc) or with 600 ppm of acrolein (4 h Acro.) under LL, on GPX5 levels. The D1 reaction centre of photosystem II (PsbA) was used for loading control.

**Figure 3 plants-11-02155-f003:**
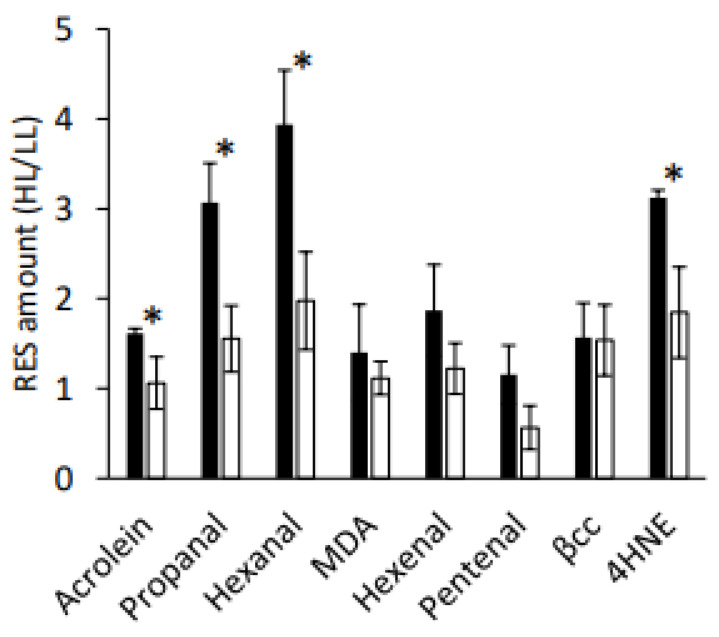
Influence of GPX5 overexpression on aldehyde/RES accumulation. Change in RES concentrations in response to HL (600 μmol photons m^−2^ s^−1^) for 4 h, relative to levels before treatment at LL (50 μmol photons m^−2^ s^−1^), in wild-type (black) and *GPXHOX-11* (white). MDA: malondialdehyde, βcc: β-cyclocitral, 4HNE: 4-hydroxynonenal. *n* = 4 ± SD with * denoting significant differences between genotypes (*p* < 0.05).

**Figure 4 plants-11-02155-f004:**
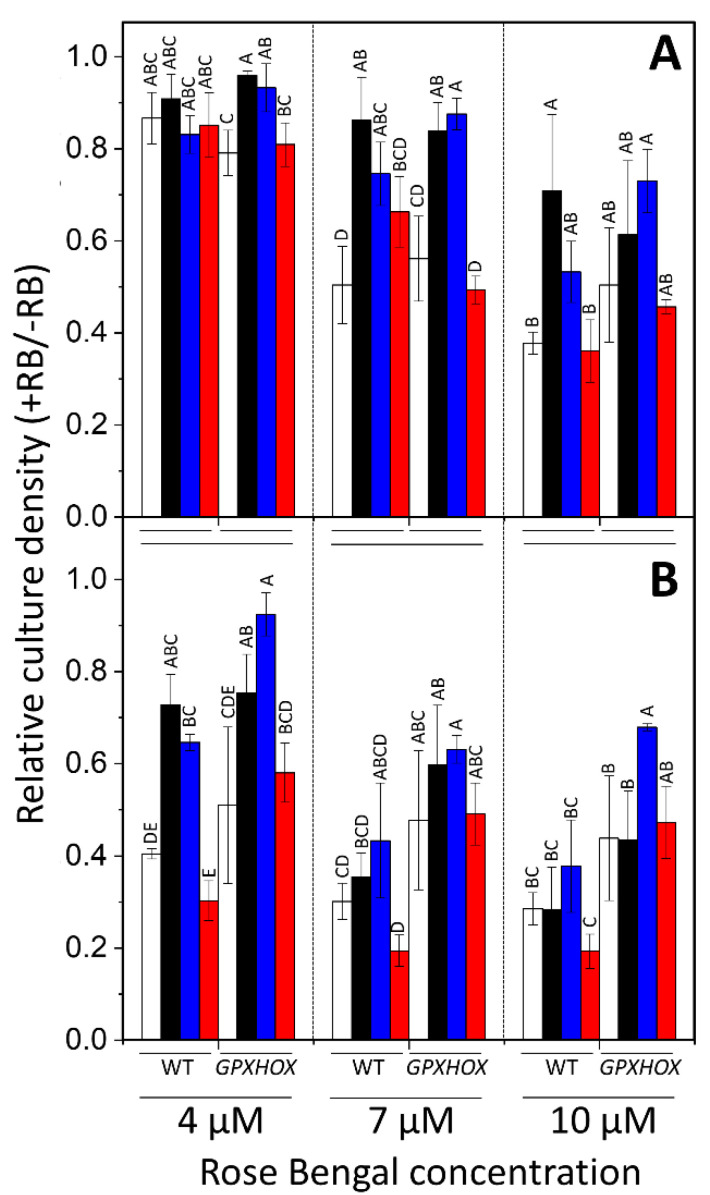
Influence of *GPX5*-overexpression and effect of HL, β-cyclocitral, and acrolein pre-treatments on tolerance to ^1^O_2_. Tolerance to ^1^O_2_ was tested by exposing WT 4A or *GPXOX-11* cultures to (**A**) 250 μmol photons m^−2^ s^−1^ for 10 min or (**B**) 50 μmol photons m^−2^ s^−1^ for 24 h, in a medium that initially contained 4 μM, 7 μM, or 10 μM of rose bengal (RB). Before treatment with RB, cells were from LL (control; white) or pre-treated with 4 h of HL (black), 600 ppm of acrolein (blue) or 600 ppm of β-cyclocitral (red) under LL. To reveal tolerance to ^1^O_2_, cell density was monitored for 24 h after RB treatment by turbidity of culture at 650 nm and is shown relative to turbidity of the respective pre-treated cultures not treated with RB. Bars labelled with different letters within the same RB treatment denote significant differences from each other (Tukey’s post hoc after arcsine transformation, *p* < 0.05, *n* = 3 ± SD).

**Figure 5 plants-11-02155-f005:**
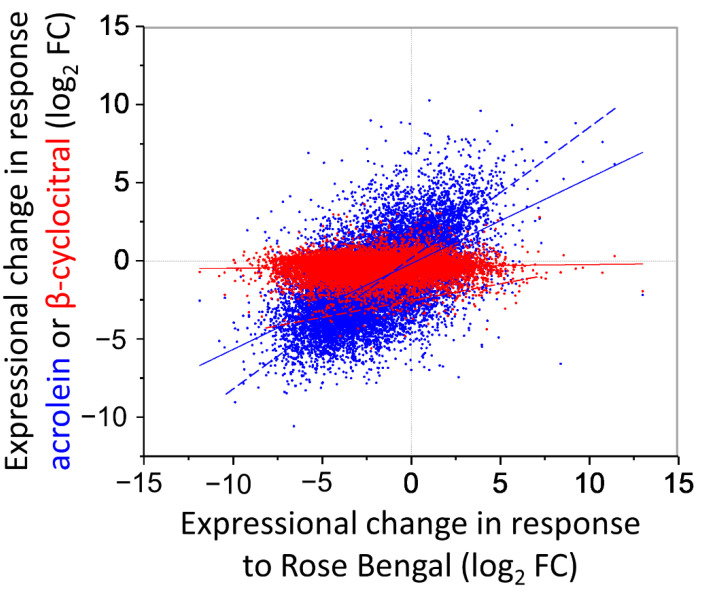
Correlation of differential gene expression induced by treatment with ^1^O_2_ and acrolein (blue) or ^1^O_2_ and β-cyclocitral (red). Levels of mRNA are expressed as log_2_ fold changes (log_2_FC), relative to mock-treated cells, calculated from RNA-Seq analyses; *n* = 3 for treatments and controls. Treatments with acrolein and β-cyclocitral were at 600 ppm and ^1^O_2_ was induced by rose bengal (RB; 1 μM under growth light at 50 μmol quanta m^−2^ s^−1^). Continuous and dashed lines of best fit consider all and only significantly affected genes, respectively. See Appendix A for expression levels of each gene. RNA-Seq data for acrolein and RB treatments adapted with permission from [13] 2018, Copyright Elsevier.

## Data Availability

Raw data is available from the corresponding author upon request.

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
