# Peer review of "β-Cyclocitral Does Not Contribute to Singlet Oxygen-Signalling in Algae, but May Down-Regulate Chlorophyll Synthesis"

_plants, 2022, doi:10.3390/plants11162155_

Round 1

Reviewer 1 Report

The paper entitled “β-cyclocitral does not contribute to singlet oxygen-signaling in algae, but may down-regulate chlorophyll synthesis” deals with the investigation of the involvement of the β-cyclocitral, a 1O2-derived breakdown product of β-carotene, in the 1O2 retrograde signaling in the microalga Chlamydomonas reinhardtii. The manuscript is well written and deals with an interesting topic but needs some improvements to be accepted for publication on Plants journal.

Following are my comments to the manuscript:

- Lines 56- 60: authors describe response to acrolein, which has not been introduced yet at this point of the manuscript (they say what is acrolein later in the text at line 70). I believe it would be better to first introduce what is acrolein and its involvement in retrograde signaling, and then describe the Chlamy’s response to acrolein.

- Authors assume β-cyclocitral could enter cells due to its influence on chlorophyll fluorescence, but is this effect established? If so, please describe it. I’m wondering if authors can provide any evidence that the compound enters the cells, maybe by HPLC or any assay able to detect the molecule?

- It is not clear why authors compared differential gene expression between β-cyclocitral and rose Bengal exposed cells (lines 68-69). Do they suppose these compounds may have the same effect? Please provide more detailed explanation of the scientific question behind this choice, the hypothesis, and the experimental plan. It is also not clear from the supplementary table 1, if the UP- and DOWN- regulated genes in Rose Bengal treatment are to be intended after also pre-treatment with β-cyclocitral or they (β-cyclocitral and rose Bengal) are individual treatments to the two substances alone. Please describe better.

- As authors write in the discussion section, in A. thaliana is known that β-cyclocitral effects the transcription of genes responsive to 1O2. Which are these genes? and do they result affected also in the present manuscript? It would be helpful to generate some plots showing the functional categories significantly enriched in genes up- and down-regulated resulted by the RNAseq analysis following β-cyclocitral treatment.

- Authors conclude that “β-cyclocitral is not a particularly reactive aldehyde since it does not lead to loss of Fv/Fm”. However, this may be due to the concentrations used, and, anyway, if the effect under study is the involvement of β-cyclocitral in the retrograde signaling, then the fact that is not toxic is something that makes sense because signaling molecules work at very low concentrations and do not have to be toxic, generally.

Author Response

We thank the reviewer for the helpful insights, comments and suggestions. Our RESPONSES are given below each point raised:

- Lines 56- 60: authors describe response to acrolein, which has not been introduced yet at this point of the manuscript (they say what is acrolein later in the text at line 70). I believe it would be better to first introduce what is acrolein and its involvement in retrograde signaling, and then describe the Chlamy’s response to acrolein.

RESPONSE: Good point! We have added to lines 49-51 "One of the most abundant RES produced by lipid peroxidation in chloroplasts due to high light stress is acrolein [10,11], which can activate a significant proportion of transcriptional changes that occur in response to 1O2 [12,13]." 

- Authors assume β-cyclocitral could enter cells due to its influence on chlorophyll fluorescence, but is this effect established? If so, please describe it. I’m wondering if authors can provide any evidence that the compound enters the cells, maybe by HPLC or any assay able to detect the molecule?

Unfortunately, due to the limited literature on physiological effects of β-cyclocitral, the changes on chlorophyll fluorescence are not so far established. However, since there is no difference between treatments with different concentrations of β-cyclocitral, except the concentration of β-cyclocitral, we deduced that β-cyclocitral must have entered the chloroplast and was causing the change in Fv/Fm and NPQ.  For confirmation, we have since included data on cellular β-cyclocitral concentrations (new Fig. S2) and added the following to results (Lines. 92-94) "Cellular concentrations of β-cyclocitral before treatment were 0.2 nmol g-1 fresh weight, and increased 400-fold 2h after exogenous treatment at 600ppm (Fig. S1)." 

- It is not clear why authors compared differential gene expression between β-cyclocitral and rose Bengal exposed cells (lines 68-69). Do they suppose these compounds may have the same effect? Please provide more detailed explanation of the scientific question behind this choice, the hypothesis, and the experimental plan.

RESPOSNE: The reason for this experiment is to see if β-cyclocitral may have a role in singlet oxygen signalling in Chlamydomonas, as has been shown in Arabidopsis (Introduction: lines 40-43). We have rewritten the text (lines 176-178) to better explain in the results the reason for comparing transcriptional response of cells treated with β-cyclocitral and with Rose Bengal (photosensitzer that produces singlet oxygen).

- It is also not clear from the supplementary table 1, if the UP- and DOWN- regulated genes in Rose Bengal treatment are to be intended after also pre-treatment with β-cyclocitral or they (β-cyclocitral and rose Bengal) are individual treatments to the two substances alone. Please describe better.

RESPOSNE: We have also amended the description of Table S1 to: "'TRUE' in columns 'U' and 'V' show if expression of the same gene was up-regulated (UR) or down-regulated (DR), respectively, by a separate treatment with Rose Bengal (RB) (>2 FC, P<0.01), as published by Roach et al., (2018). 

- As authors write in the discussion section, in A. thaliana is known that β-cyclocitral effects the transcription of genes responsive to 1O2. Which are these genes? and do they result affected also in the present manuscript? It would be helpful to generate some plots showing the functional categories significantly enriched in genes up- and down-regulated resulted by the RNA seq analysis following β-cyclocitral treatment.

RESPONSE: This is a good point. We checked the literature and found a few genes that could also be related to chlorophyll anabolism affected by β-cyclocitral in Arabidopsis. We added to the discussion (Lines 277-281) "In A. thaliana, β-cyclocitral also down-regulated expression of CHLI2 and a few other genes involved in porphyrin/chlorophyll biosynthesis, such as PORB, CHLM and HEME1 alongside an up-regulation of CLH1 and CLH2 involved in chlorophyll catabolism (Ramel et al., 2102), indicating a conserved signalling role for β-cyclocitral in decreasing chlorophyll contents, which pre-existed before evolution of vascular plants." Regarding the second suggestion of including functional categories significantly enriched in genes up- and down-regulated following β-cyclocitral treatment, there is only one pathway with more than one KEGG ontology hit, and that was ‘porphyrin and chlorophyll metabolism’, as stated on line 186 and already shown in Fig. S5.

- Authors conclude that “β-cyclocitral is not a particularly reactive aldehyde since it does not lead to loss of Fv/Fm”. However, this may be due to the concentrations used, and, anyway, if the effect under study is the involvement of β-cyclocitral in the retrograde signaling, then the fact that is not toxic is something that makes sense because signaling molecules work at very low concentrations and do not have to be toxic, generally.

RESPONSE: To see any response of β-cyclocitral on Chlamydomonas we had to use a minimum treatment concentration of 600ppm (Fig. S1), despite that this is likely higher than physiological levels (Fig. S2). This was also selected because it is the same concentration of treatment used for acrolein so we could make a cross comparison between molecules. We have added to the discussion (lines 220-222) "The concentration of β-cyclocitral even in very light-stressed C. reinhardtii has never been measured >1 nmol g-1 FW [25], which is below the cellular concentrations after exogenous treatment with 600ppm (Fig. S2). Therefore, our data supports that unlike acrolein, β-cyclocitral does not build to toxic concentrations in light-stressed cells."

Reviewer 2 Report

The authors prepared a rather condensed manuscript, which was not easy to follow up and understand, in particular for those who might not be directly involved in the analysis of stress-responses in green algae, which were compromised by various agents in complex pharmacological experimental approaches. Although the authors present several confirmative results, I would have appreciated to read a more detailed and comprehensive description of experiments combined with a more extensive explanation of the rationales for these experiments. Together with clearly described results this manuscript would certainly gain an increasing visibility and an improved acknowledgement of readers can be assured. I assume that this could be true, although the main messages derived from the presented data only indicate a negative outcome that β-cyclocitral did not affect a protective signaling in Chlamydomonas on different 1O2-generating stress conditions, unlike for example other electrophile compounds.

 I suggest to improve readability with additional clarifications.

 -      In Figure 1 it is not obvious which concentration of β-cyclocitral was applied. It is also not clear how a correlation between the applied amounts of β-cyclocitral and 50.000 ppm atmospheric concentration can be built up.

-      Are the GPX-overexpressor lines generated in the wild type line 4A background?

-      Figure 1. It is not obvious whether the strains were incubated in LL or HL. A more detailed legend could give more details in these disclosures.

-      Page 4. As the description of the complex experiments of Figure 4 is very short, it will help when at least the description contains more information, what has been compared with what to deduce any essential statement. Then it can become clearer which results are most relevant.

-      As it can become obvious that only a few aspects of results in Figure 4 have been selected, it is not automatically evident to generalize the statements. “Relative to levels before treatment” means that these values were 1, weren’t they? Paragraph on page 4. Are comparisons described relative to LL conditions of the pretreatment?

-      It would have been nice when the RNA-seq data on those genes with a significant deregulation of their expression would have been confirmed by separate quantitative transcript analysis (qRT-PCR).

Author Response

We thank the reviewer for the helpful insights, comments and suggestions. Our RESPONSES are given below each point raised:

-The authors prepared a rather condensed manuscript, which was not easy to follow up and understand, in particular for those who might not be directly involved in the analysis of stress-responses in green algae, which were compromised by various agents in complex pharmacological experimental approaches. Although the authors present several confirmative results, I would have appreciated to read a more detailed and comprehensive description of experiments combined with a more extensive explanation of the rationales for these experiments.

RESPONSE: We agree that text was too concise and apologise for the difficulty you had in understanding the rationale behind the experiments and overall lack of clarity. We have modified the summary of experimental approach in the introduction (lines 72-81), and to each of the sentences introducing the results of each Figure (Line 99 for Fig S3;  lines 114-116 for Fig. 2; Lines 146-150 for Fig. 4; Lines 176-179 for Fig. 5).  

-Together with clearly described results this manuscript would certainly gain an increasing visibility and an improved acknowledgement of readers can be assured. I assume that this could be true, although the main messages derived from the presented data only indicate a negative outcome that β-cyclocitral did not affect a protective signaling in Chlamydomonas on different 1O2-generating stress conditions, unlike for example other electrophile compounds.

RESPONSE: We have since focused more on the down-regulation of chlorophyll synthesis in the abstract (lines 21-22) and discussion (lines 277-282), since this was also found to occur in plants, indicating an early origin of this pathway.  However, we believe since β-cyclocitral has now such an established role in singlet oxygen signalling in plants, showing this is not the case in algae is a valuable insight, albeit being a 'negative' result in the submission.

 -      In Figure 1 it is not obvious which concentration of β-cyclocitral was applied. It is also not clear how a correlation between the applied amounts of β-cyclocitral and 50.000 ppm atmospheric concentration can be built up.

RESPONSE: We added the ppm concentration to the axis label in Fig. 1 and Fig. S1, and the calculation of ppm in the methods section was elaborated (Lines 321-324)

-  Are the GPX-overexpressor lines generated in the wild type line 4A background?

RESPONSE: Yes. Information added in results (Line 117) and in Methods (Line 295).

-      Page 4. As the description of the complex experiments of Figure 4 is very short, it will help when at least the description contains more information, what has been compared with what to deduce any essential statement. Then it can become clearer which results are most relevant.

RESPONSE: We have extended the description in the legend of Fig. 4 and the text in the results (lines 146-154) to make sure it is clear why and how the treatments were made, and  what exactly has been compared in the analysis of results (lines 156-164), and also provided a summarising statement to leave the reader with a clear message.

- As it can become obvious that only a few aspects of results in Figure 4 have been selected, it is not automatically evident to generalize the statements. “Relative to levels before treatment” means that these values were 1, weren’t they? Paragraph on page 4. Are comparisons described relative to LL conditions of the pretreatment?

RESPONSE: We modified the figure legend of Fig. 4 to clarify that data of culture turbidity (i.e. cell density), as used as a proxy for singlet-oxygen tolerance, is relative to cell growth in absence of Rose Bengal. It now reads "To reveal tolerance to 1O2, cell density was monitored 24h after RB treatment by turbidity of culture at 650 nm, and is shown relative to turbidity of the respective pre-treated cultures not treated with RB." For statistical analyses comparisons were made to LL-treated (control) cells, and reference to "before treatment" has been deleted to avoid confusion. 

-      It would have been nice when the RNA-seq data on those genes with a significant deregulation of their expression would have been confirmed by separate quantitative transcript analysis (qRT-PCR).

RESPONSE: Yes, we agree, but unfortunately are unable to perform this experiment in our labs.

Reviewer 3 Report

The responses of Chlamydomonas reinhardtii to carotene oxidation product, beta-cyclocitral were compared to high light responses. This study focused on singlet oxygen stress and GPX5. The over-expression of GPX5 was used. The effects of beta-cyclocitral on GPX5 expression (protein) were examined. The authors found that beta-cyclocitral and acrolein did not affect the GPH expression.

The major comment is that the relationship of transcriptomic data in response to beta-cyclocitral with NPQ is not discussed and it seems it is not associated with the NPQ. 

Minor comments

Lines 65-71: because the materials and methods are described in the last paragraph of the manuscript, the materials regarding the strains used in this study and their characteristics are better described in more detail in the Introduction for the readers.

Line 76: the full name of WT-4 when first mentioned

Line 91: over-expression strain is generally abbreviated as OE

It cannot be accepted by publication.  

Author Response

We thank the reviewer for the comments and suggestions. Our RESPONSES are given below each point raised:

The major comment is that the relationship of transcriptomic data in response to beta-cyclocitral with NPQ is not discussed and it seems it is not associated with the NPQ. 

RESPONSE: The effect on NPQ was also found in various NPQ mutants knocked out for specific NPQ pathways, hinting at a direct effect on the thylakoid luminal pH that activates NPQ (e.g. similar to the behaviour of nigericin or an uncoupler like FCCP), rather than a process involving a change in transcription for a protein involved NPQ. We modified the results (line 98) and discussion to clarify this point (lines  227-232) to "Inhibition of NPQ in various NPQ mutants indicates that β-cyclocitral directly affected an over-riding NPQ mechanism, such as the requirement of low luminal pH. Less accumulation of zeaxanthin under HL, which requires a low luminal pH for violaxanthin de-epoxidase activity for production, supported this (Fig. S1). Therefore, we suggest that β-cyclocitral may act as an uncoupler of the thylakoid membrane potential, comparable to the inhibitory activity of structurally similar monoterpene ketones, such as pulegone, on respiration [24]."

-Lines 65-71: because the materials and methods are described in the last paragraph of the manuscript, the materials regarding the strains used in this study and their characteristics are better described in more detail in the Introduction for the readers.

RESPONSE: Good point! We have since introduced the GPX5 overexpressor in the introduction (lines 64-65), and the wild types in Results on line 87. The wild-type background of GPX5 overexpressors has also been given in Results on line 117. The finer details on strain numbers has been left in the Methods and Fig. S1. 

Line 76: the full name of WT-4 when first mentioned

RESPONSE: added

Line 91: over-expression strain is generally abbreviated as OE

RESPONSE: We agree, but for keeping consistency with the name of the strains originally published, we have stuck with OX. 

Round 2

Reviewer 1 Report

For me the revised form is fine to be published in Plants journal.

Author Response

thankyou

Reviewer 2 Report

The authors took an effort to improve the manuscript. It was intended to simplify the written text, to provide more explanations for the design of experiments, to relativize and qualify certain statements. However, at certain text passages the coherence still can be improved and the statements should be adjusted to be more consistent with the results of the findings.

I initially obtained a revised manuscript version with track changes. Reading was quite confusing and I hope the authors themselves had no difficulties to following the final revised version. I found that this version also differs from a word file containing another revised version, which was provided under down load unpublished data? I hope that I considered the final resubmitted and revised version.

Abstract: Compare the two redundant sentences of lines 17 and 20, which should be combined. But what is the basis for this assertion that inactivated synthesis of chlorophyll indicates a conserved

The authors often refer to their previous paper with reference 13 “Roach, T.; Stoggl, W.; Baur, T.; Kranner, I. Distress and eustress of reactive electrophiles and 451 relevance to light stress acclimation via stimulation of thiol/disulphide-based redox defences. Free 452 Radic. Biol. Med. 2018, 122, 65-73”. This previous paper contains likely many indications and suggestions, which were picked up again here and it is not entirely clear to which extent sufficient novelty can be ensured in the recent manuscript, inter alia the RNA seq data, which were already published in Ref. 13. Following this line, was not sure if the numbers in the manuscript (line 205ff) or in Fig.S4 should count, when the authors report about common genes up-regulated by Rose Bengal and beta-cyclocitral.

Author Response

Thank you for your assessment of our resubmission and suggestions for improvement. We have made some minor changes in the order of the text and broken up two long sentences to further improve clarity.

RESPONSE TO REVIEWER:

Abstract: Compare the two redundant sentences of lines 17 and 20, which should be combined. But what is the basis for this assertion that inactivated synthesis of chlorophyll indicates a conserved.

RESPONSE: Thanks for spotting this. Unfortunately, we cannot combine the two sentences because, although the genes are partly homologous, are different between Arabidopsis and Chlamydomonas. Therefore, we modified the second sentence to refer to "this pathway" and avoid the redundancy of saying "porphyrin and chlorophyll anabolism" twice. The use of 'conservation' is because down-regulation of genes related to chlorophyll anabolism in rsposne to beta-cyclocitral occurs in algae and plants, thus before the split of Chlorophytes and Streptophytes.

The authors often refer to their previous paper with reference 13 “Roach, T.; Stoggl, W.; Baur, T.; Kranner, I. Distress and eustress of reactive electrophiles and 451 relevance to light stress acclimation via stimulation of thiol/disulphide-based redox defences. Free 452 Radic. Biol. Med. 2018, 122, 65-73”. This previous paper contains likely many indications and suggestions, which were picked up again here and it is not entirely clear to which extent sufficient novelty can be ensured in the recent manuscript, inter alia the RNA seq data, which were already published in Ref. 13. Following this line, was not sure if the numbers in the manuscript (line 205ff) or in Fig.S4 should count, when the authors report about common genes up-regulated by Rose Bengal and beta-cyclocitral. 

RESPONSE: The RNA Seq. data of beta-cyclocitral-treated cells has not been published before. It is reasonable to compare this data to previously published data sets (e.g. Fig. 5; Fig. S4), as long as this has been clearly referenced to the original work, as has been done. This forms the basis for answering the research question if beta-cyclocitral has a role in singlet oxygen signalling in Chlamydomonas.

Reviewer 3 Report

I have found that most of the comments have been considered in the revised manuscript.

Accordingly, it is suggested it can be accepted for publication in the present form.

Author Response

thankyou